# The Role of Selected Speech Signal Characteristics in Discriminating Unipolar and Bipolar Disorders

**DOI:** 10.3390/s24144721

**Published:** 2024-07-20

**Authors:** Dorota Kamińska, Olga Kamińska, Małgorzata Sochacka, Marlena Sokół-Szawłowska

**Affiliations:** 1Institute of Mechatronics and Information Systems, Lodz University of Technology, 116 Żeromskiego Street, 90-924 Lodz, Poland; 2Systems Research Institute, Polish Academy of Sciences, 01-447 Warsaw, Poland; o.kaminska@ibspan.waw.pl; 3Britenet MED Sp. z o. o., 00-024 Warsaw, Poland; malgorzata.sochacka@britenet.eu; 4Outpatient Psychiatric Clinic, Institute of Psychiatry and Neurology, 9 Jana III Sobieskiego Street, 02-957 Warsaw, Poland; msokol@ipin.edu.pl

**Keywords:** bipolar disorder, depression, mania, speech signal, classification, machine learning, healthcare application

## Abstract

Objective:The objective of this study is to explore and enhance the diagnostic process of unipolar and bipolar disorders. The primary focus is on leveraging automated processes to improve the accuracy and accessibility of diagnosis. The study aims to introduce an audio corpus collected from patients diagnosed with these disorders, annotated using the Clinical Global Impressions Scale (CGI) by psychiatrists. Methods and procedures: Traditional diagnostic methods rely on the clinician’s expertise and consideration of co-existing mental disorders. However, this study proposes the implementation of automated processes in the diagnosis, providing quantitative measures and enabling prolonged observation of patients. The paper introduces a speech signal pipeline for CGI state classification, with a specific focus on selecting the most discriminative features. Acoustic features such as prosodies, MFCC, and LPC coefficients are examined in the study. The classification process utilizes common machine learning methods. Results: The results of the study indicate promising outcomes for the automated diagnosis of bipolar and unipolar disorders using the proposed speech signal pipeline. The audio corpus annotated with CGI by psychiatrists achieved a classification accuracy of 95% for the two-class classification. For the four- and seven-class classifications, the results were 77.3% and 73%, respectively, demonstrating the potential of the developed method in distinguishing different states of the disorders.

## 1. Introduction

This article focuses on the two most common affective disorders: bipolar disorder (BD) and unipolar disorder (UD). Both are chronic, recurrent, and highly morbid mental illnesses [1]. According to the ICD-10 classification, affective disorders are classified into manic episode, bipolar disorder, depressive episode, recurrent depressive disorder (unipolar disorder), persistent mood disorders (cyclothymia, dysthymia), other mood disorders, mood disorders, and not specified [2].

Unipolar disorder is a mental illness characterized by recurring depression episodes, and bipolar disorder by recurring episodes of depression and mania. According to the report of the Organization for Economic Co-operation and Development and the European Commission, *Health at a Glance: Europe 2018* [3], more than one in six people in EU countries (17.3%) had a mental health problem in 2016, of which 21 million Europeans (4.5%) had depression or depressive states, and bipolar disorder affected nearly 5 million people (1% of the population) [4]. Subsequent episodes seem to worsen the prognosis and increase the risk of suicide [5,6]. Patients in the initial stage of bipolar disorder respond better to treatment; therefore, early intervention strategies may be vital to improving disease outcomes by reducing the conversion rate to full-blown disease and the severity of symptoms [7,8].

The main BD and UD treatment is aimed at reducing the current symptoms during an episode and preventing episodes at the same time by combining the following:Pharmacotherapy—the mood is controlled (i.e., stabilized), and symptoms are alleviated with the help of an adapted and difficult-to-define combination of at least one of the following: antidepressants, antipsychotics, mood stabilizers, and other drugs, e.g., anxiolytics, hypnotics.Psychotherapy—patients are trained to manage symptoms and find practical ways to prevent episodes through behavioral and lifestyle choices such as routine sleep, social activity, and appropriate stress management.Psychoeducation—patients and their relatives and close friends are taught about the intricacies of unipolar and bipolar disorders, the causes of episode recurrences, and how to deal with the disease.

One approach to treatment is anticipating and preventing episodes by training patients to recognize their early warning message symptoms collated into Early Warning System (EWS)—symptoms indicating an upcoming episode [9]. However, training is resource-intensive, and its success varies significantly from patient to patient. Unfortunately, some patients can never identify the patterns in their episodes that reveal the EWS. For many years, the diagnosis and treatment of BD and UD have been based on mood charts, creating daily records of states, behavior, and mood. This helped patients identify patterns and track progress [10]. In these times, mobile phones are used by the general public, which could be an opportunity for patients with affective disorders.

Web or cellular solutions can make it easier for patients to report data and reduce data inconsistency by guiding data input. However, existing websites and mobile applications often lack a clinical utility perspective and generalization for different patients. An objective, individualized approach to the symptoms of BD and UD with the use of artificial intelligence can be a technological breakthrough in the diagnosis and effective treatment of recurrent episodes of affective disorders in the future. The use of AI can alleviate problems related to the objectivity of data obtained through the self-observation of patients or their observation by family or friends and offer observation quality comparable to that provided by professionals while assuring otherwise unattainable continuity at an affordable cost, resulting in earlier diagnosis of a relapse and treatment adjustment at an earlier stage, and thus improving its results.

The research by Nicholas et al. [11] showed that the analytical solutions (which met the minimum quality standards) did not enable active tracking of patients’ behavior and identifying episodes of affective disorders based on this information. Since then, individual reports have appeared in the literature on the studied solutions using AI and based on passive monitoring of patients’ behavioral parameters (more about them in the section on the novelty of the project results). They confirm the potential for using machine learning methods to monitor mental health based on continuous sensory data. However, the set of analyzed parameters differed each time, and their number was limited.

In this study, we utilized techniques in speech signal processing and machine learning to identify bipolar and unipolar disorders based on short audio recordings. We gathered a corpus from 100 diagnosed patients, which psychiatrists annotated during regular visits. We explored various methods for selecting features and performing classification and provide a detailed description of robust baseline models. The main contributions of this research include the creation of a novel audio corpus for unipolar and bipolar disorders, along with comparative experimental results that evaluate different approaches to manage this issue.

The rest of the paper is organized as follows. Section 2 discusses the previous work on depression and mania recognition based on speech, including datasets, features, and most commonly used classifiers. Section 3 presents the experimental specification and explains the selection of features used in our study. Section 4 presents the results of our experiments. We discuss our findings in Section 5 and provide some final remarks.

## 2. Related Research

Moods can be expressed through different modalities. There is evidence that the affective state of individuals is correlated with facial expressions [12], body language [13], voice [14], and physiological changes [15]. Since mental disorders affect one’ s mood, it has been proven that the signals mentioned above can be used for unipolar [16,17,18] and bipolar [19,20,21] detection. However, since, in this study, we sought to determine the effect of a specific disease on the speech signal, in this section, only speech-based state-of-the-art research is presented. We discuss the approach to the experiment and creation of a speech database, the most commonly used speech parameters, and recognition methods.

### 2.1. Speech-Based Depression Recognition

#### 2.1.1. Datasets

In contrast to affect recognition, where acted, evoked, and natural speech can be used, depression datasets are mainly created during conversations between a clinical doctor and depressed patients or through telephone/virtual interviews. The most popular, publicly available depressive speech datasets are summarized in Table 1.

The AVEC2014 [22] dataset is the Audio–Visual Depression Language Corpus subset of 300 videos ranging from 6 s to around 4 min. The recordings were gathered during human–computer interaction scenarios: while reading aloud and responding to different questions in German. The recordings were labeled into dimensions of Valence, Arousal, and Dominance (VAD) and self-reported Beck Depression Index-II (BDI-II).

Wizard of Oz (DAIC-WOZ) [23] is a subset of Distress Analysis Interview Corpus, which contains clinical interviews between computer agents (controlled by a human interviewer) and patients diagnosed with psychological distress conditions such as anxiety, depression, and post-traumatic stress disorder. The whole dataset consists of 189 sessions ranging between 7 and 33 min (average is 16 min). In addition to audio–video recordings, the corpus also contains information on galvanic skin response, electrocardiogram, and participants’ respiratory data. Its extension is EDAIC (Extended DAIC) database [24]. Both are freely available for research from The University of Southern California Institute for Creative Technologies website.

MODMA [25] is a Chinese open dataset that includes data from 23 clinically depressed patients (diagnosed and selected by professional psychiatrists) and 29 healthy control subjects aged 18–52. Speech samples were recorded during interviewing, reading aloud, and picture description.

As one can quickly notice that public depression databases available for research are limited and far from the scientific research needs. The unavailability of such corpora is undoubtedly due to the sensitivity of depressive disorder and ethical issues. Consequently, most institutions could not obtain sufficient samples or cannot make them available.

#### 2.1.2. Depression Recognition Methods

There are two common approaches to speech signal processing. The first one (see Figure 1A) is based on acoustic features related to the topic of concern and uses traditional machine learning algorithms as classifiers. In the case of speech-based depression recognition, the most commonly used acoustic features are

Prosodies such as pitch, fundamental frequency (F0), energy, speaking rate, and pauses [26,27,28];Spectral features like MFCC [29,30,31] and LPC [32,33];Vocal tract descriptors—formants [34,35];Voice quality features, e.g., jitter and shimmer [36].

In this approach, after feature extraction, traditional classification such as Support Vector Machine (SVM) [36], Logistic Regression (LR) [37], Decision Tree [25], Gaussian Mixture Model (GMM) [38], etc., is applied. Recently, this conventional approach has been replaced by deep learning methods (see Figure 1B). There are two different options for using deep learning in speech processing. First, traditional acoustic features can be used to train deep classifiers, which will then be able to recognize or predict. Alternatively, a raw signal or spectrogram can be pushed into a deep architecture, which can learn high-level features by itself [39]. For this purpose, convolutional neural network (CNN) [40], long short-term memory (LSTM) [41], or recurrent neural network (RNN) [42] structures are usually used.

### 2.2. Speech-Based Mania Recognition

#### 2.2.1. Datasets

In contrast to speech depression recognition, the number of papers examining mania is limited. Additionally, in this case, only one database is available (according to our knowledge) for the research of maniac speech. The Bipolar Disorder corpus [43] was made available for research purposes as part of the 2018 Audio/Visual Emotion Challenge [44]. The corpus consists of 218 speech samples gathered among 46 patients and 49 healthy controls. The samples were annotated for BD state as well as the Young Mania Rating Scale (YMRS) by psychiatrists. The BD-diagnosed individuals were asked to explain the reason for coming to the hospital/participating in the activity, describe happy and sad memories, count up to thirty, and describe two emotion-eliciting pictures. In addition to the tasks mentioned above, the control individuals were asked to act out mania and depression conditions.

Despite the lack of available datasets, which may be related to ethical issues, the research on mania recognition based on speech signals is ongoing. Most of the research is based on private speech samples collected among diagnosed patients via phones equipped with an automatic voice-collecting system [45] or clinical interviews [46].

For example, in [45], smartphones equipped with a voice-collecting system software were used to gather the baseline from 30 manic patients. The recordings, in the form of a conversation with a psychiatrist, took place in a quiet room with surrounding noise under 30 dB. After the conversation, the psychiatrist assessed the clinical symptoms of the interlocutors using BRMS and Clinical Global Impression (CGI).

In [47], the authors gathered spontaneous speech samples via modern smartphone devices among 21 hospitalized patients diagnosed with bipolar disorder and in a manic episode upon admission to the hospital. Similarly, BRMS was used to assess the patients by a psychiatrist for determining manic state.

Spontaneous speech derived from telephone calls was used also in [48], where the authors investigated whether voice features could discriminate between BD and healthy control (HC) individuals. Speech samples were collected daily during ordinary phone calls for up to 972 days. A total of 121 patients with BD and 38 HC were included. In addition, patients were asked to evaluate their symptoms daily via a smartphone-based system.

#### 2.2.2. Mania Recognition Methods

Similar to depression recognition, several speech features have been identified as potential markers of mania. Some of these features include

Prosodies such as pitch, intensity (energy), rate of speech, and pause duration [49];Spectra-like MFCC [47] and LPC coefficients [50].

There are several classifiers that have been used as classifiers, including SVMs [47], Random Forests [51], LR [52] and Neural Networks.

## 3. Methodology

### 3.1. Experimental Specification

Our study was conducted at a clinic in Warsaw (Poland) in 2021–2022 as a part of the observational clinical trial, where we obtained data from 100 patients (60 women and 40 men). As for the disease distribution, monitored patients were categorized as bipolar (75–75% of all patients) regardless of the type (type I or II) and unipolar (25–25% of all patients). In order to ensure a high number of observed changes, only patients with a relatively rapid phase change were recruited: the inclusion criteria required at least 2 phase changes during the last 12 months in bipolar patients and at least one episode during the last 12 months in unipolar patients.

During the study, patients used a dedicated smartphone application enabling them to provide daily voice samples. The application was installed on each patient’s phone. Phones with Android version over 6.0 were compatible.

The voice sample was gathered in the form of a recording of answers to 3 randomly selected questions from a list of 85 emotionally indifferent questions, which did not refer to the disease or other sensitive topics. An example of the question: “Which time of the year do you like most and why?”. The time of day to record the sample was irrelevant and patients were instructed to select the time and place of their choice possibly assuring the least environmental noise. The length of the recording was demonstrated to the patient on the phone display during the recording, and patients were encouraged to provide samples at least 30 s long. Samples longer than 3 min were trimmed to spare phone memory. Questions were not repeated during the study due to their limited number. However, patients were informed that the content of the answer was of no importance and did not need to be related to the question. In total, 12,400 voice samples were gathered during the study. Samples were not processed on the phone, and usage of special phone functions related to the recording, like ANC, was not controlled. Raw data were forwarded to the central system for processing. No control of recording quality was applied in order to keep trial conditions as similar as possible to the conditions of real life usage.

The study was designed following the methodology typical for this kind of trials, and no healthy controls were included. For each patient, the control was his/her own behavior in remission (euthymia).

During the study, patients were regularly examined by a psychiatrist, and their mental state was assessed with a CGI scale, thus labeling their mood. Personal visit was happening every 3 months and telephone contact every two weeks.

A graphical summary of the distribution of patients by gender and disease entity is presented below Figure 2.

Concerning the age of the individual patients, the distribution of the population included in the observational part of the clinical trial is as follows Figure 3:

The average age of the participants in the study was 41 years, while the youngest patient was 20 years old, and the oldest one was 66 years old—the median was 41.

In designing the clinical trial, we assumed that the minimal number of patients needed for model development was 100 patients that meet the inclusion criteria (e.g., age or no hearing disability as the trial required voice recording). It was assumed that at least 75 patients with bipolar disorder and 25 patients with depression (unipolar disorder) would be acquired.

A total of 546 changes in the mental states were observed in the monitored patients. Changes in the mental states were related to observations for which, for two subsequent visits, the patient was assigned a different state relative to the CGI scale. An important issue related to the obtained labels is the fact that the labels are known only on days with medical appointments (i.e., when the psychiatric doctor assigns such a label); there were 1902 visits to physicians (i.e., physical or telephone visits) during the clinical trial (on average 19 psychiatric appointments per patient). However, in the case of patients whose mental state with regards to the CGI scale had not changed between two consecutive medical visits, the status label from the visit days was assigned for all days between these visits—all observations with labels (i.e., also with the indicated supplement) were 4662. The frequency of contacts with patients and frequency of labeling was adjusted to the ICD-10 standard: either personal or phone visit happened every 14 days in order to ensure that labels extended to days between visits were still valid.

The scope of attributes collected in the observational part of the clinical trial included

Patient voice data related to the daily recordings of the voice samples.Sleep data (i.e., Mi Fit band or sleep survey).Data on physical activity (e.g., number of steps).Other behavioral data (e.g., number of phone calls made).

From a process perspective, sleep data, physical activity data, and other behavioral data required little patient involvement—they were largely collected automatically (except for the sleep questionnaire completed manually by patients). By far the greatest involvement of patients was related to the voice data—every day, patients received notifications about the need to record a voice sample (i.e., answer a randomly selected question). Therefore, the greatest shortcomings in terms of data appeared in the area of voice sample recordings—the percentage distribution of the number of patient recordings is presented below Figure 4.

On average, there were 124 recordings per patient; there were also individual cases of patients with no recordings or with a small number of recordings. Such cases were individually discussed with doctors or patients.

The key information related to the patient data obtained is summarized below:The observational part of the clinical trial involved the observation of 100 patients (60 women, 40 men);A total of 546 changes in patients’ condition were observed during this period;The average age of the patients in the study was 41 years; most of them were single (48), married (37), divorced (13), or widowed (2);On average, each patient had 124 recordings of voice samples.

Consent for clinical trials was obtained from the Office for Registration of Medicinal Products, Medical Devices, and Biocidal Products (Poland) on 25 May 2021 (decision number UR.D.WM.DNB.39.2021).

### 3.2. Speech Signal

Theoretical and practical approaches to affective computing suggest that specific patterns of vocal modulation are correlated with a particular vocal affect. Emotions and moods may cause modifications in breathing patterns, phonation, or articulation, which are reflected in the speech signal [53]. Early studies found that even simple features like pitch, speech rate, loudness, inflection, and rhythm influence emotion perception and can be used for discriminating certain affective states [54]. For example, states such as anger and fear are characterized by accelerated speech rate, high fundamental frequency (F0) values, and a wide intonation range due to sympathetic nervous system activation, rapid heart rate, and increased blood pressure, which may sometimes be accompanied by a dry mouth and muscle tremor. The opposite happens in the case of sadness and boredom. Conversely, when speech is slow and monotonous, F0 is lowered without major changes in intonation. It is caused by a stimulation of the parasympathetic nervous system, slowing heart rate, a drop in blood pressure, and increased saliva production. However, more complex patterns like linear predictive coding parameters (LPC) or Mel-frequency cepstrum coefficients (MFCC) have also shown promise for emotional state recognition [55].

Our hypothesis, based on consultations with psychiatric experts, stems from the assumption that there is a solid correlation between affective states and unipolar or bipolar disorders. Therefore, discriminative speech features in emotion recognition can be applied to distinguish between diffeerent patient states in the above-mentioned disorders. In this section, a short description, as well as the hypothesis defining the use of particular features, is presented.

#### 3.2.1. Fundamental Frequency (F0), Pitch

The frequency of the vocal folds, the so-called fundamental frequency (F0), is an individual characteristic resulting from the size of the larynx and the tension and size of the vocal cords and is gender- and age-dependent. It directly affects the voice scale. During the conversation, the extent of its changes is mainly related to intonation, which plays a huge role in expressing emotions, and that is why sound source descriptors are commonly used in research on this issue. Pitch is more often used to refer to how the fundamental frequency is perceived [56].

**Hypothesis** **1.**The intensity of emotions increases significantly in mania. The strength of emotions is related to accentuation and intonation, which is characterized by F0, the intensity and energy of the sound. As the intensity of emotion increases, the pitch range becomes much broader, with steeper slopes. It is also associated with an increase in mean F0, range, and variability. On the other hand, during the depression, grief, sadness, and slight thoughtfulness are perceptible in the voice. In the case of these emotions, there is a general decrease in the mean value of F0, its range, and its variability, and a downward direction of the intonation contour. The F0 contour is essentially constant (so-called monotone speech); the average F0 is higher and has a broader range.

#### 3.2.2. Short-Time Energy

The short-time energy (STE) describes the envelope of the speech signal, reflects changes in its amplitude, and is widely used in various aspects of speech recognition. The voiced part of speech, due to its periodicity, has high energy, and the voiceless part has low energy. Thus, loud speech and question types are characterized by higher energy. Neutral speech has the lowest energy changes. On the other hand, fast and precise speech is characterized by high energy fluctuation [57].

**Hypothesis** **2.**It is assumed that in the case of depressed speech, the signal energy will be stable, while in the case of mania, where speech is fast and loud, the fluctuation of signal energy will be significant. The standard deviation of the energy values indicates the degree of stability of the modulation, and therefore should also differentiate well between mania and depression.

#### 3.2.3. Zero Crossing Rate (ZCR)

In the context of a discrete signal, ZCR occurs if successive signal samples have different algebraic signs. Thus, ZRC is a measure that indicates how many times in a given time interval/frame the amplitude of speech signals passes through zero. In the case of emotional speech, such as joy or anger, one can speak of high arousal, while in the case of a neutral or sad state, of low arousal. Based on many studies on emotional speech, it has been determined that the average ZCR is high for emotional states of high arousal and low for those of low arousal [58].

**Hypothesis** **3.**The preliminary analysis determined that depressive states, unlike mania, are associated with low arousal. Thus, we can assume that the average ZRC will vary depending on the disease entity.

#### 3.2.4. Pauses

Pauses are temporal parameters of a speech signal that can be determined directly from its temporal structure. Analysis in the time domain is mainly used to assess prosodic properties of the voice, such as voice loudness, speech rate, and fluency [57].

**Hypothesis** **4.**According to the observation, people in mania do not use pause during speech; thus, the number of short, medium, and long pauses has been taken into consideration. Additionally, based on these parameters, one can determine the pace of speech (speeding up, slowing down).

#### 3.2.5. Jitter and Shimmer

Jitter and shimmer, representing the variations that occur in the F0, have repeatedly been proven valuable in speech signal analysis. Jitter is defined as a parameter of frequency variation from cycle to cycle. Shimmer refers to the amplitude variation of the speech signal. The value of the jitter is mainly affected by the lack of control of vocal cord vibration: the voices of patients with pathology often have a higher jitter value. Shimmer changes with decreasing glottal resistance and vocal cord modifications and is correlated with the presence of noise emission and dysphonia [59].

**Hypothesis** **5.**It is assumed that amplitude instability can be observed among patients with mania. In contrast, vocal amplitude stability (even monotonicity) is characteristic of depression. Jitter parameter, which is associated with a large amount of noise, may well characterize mania.

#### 3.2.6. LPC and Formants

Since the speech signal is formed from the signal produced by the vocal folds subjected to multiple reflections in the resonant cavities, it can be represented as the sum of several various delayed activation signals. Thus, the speech signal in a given moment is the sum of samples of that signal in previous moments. Because of its close relationship to the operation of the resonant tack, linear prediction is particularly suited to the determination of its resonant frequencies (formants) [60].

**Hypothesis** **6.**Formants characterize changes in the shape of the vocal tract and, therefore, the manner of articulation (tongue and mouth position, mandibular mobility). They are important information in cases of the so-called jawbone or crying (muscle tremor), which have been observed in sample patients. Formants may inform us about mumbling and babbling. Additionally, based on the available literature, it is safe to say that LPC coefficients have been used successfully to identify speech, emotion, and voice dysfunction states.

#### 3.2.7. MFCC

The human auditory system distinguishes low frequencies better than high frequencies. Therefore, the frequency scale expressed in Hertz is converted to a frequency scale expressed in Mel to reproduce this property. MFCC coefficient values result from filtering the cepstrum with successive filters from the Mel filter bank. The MFCC coefficients of lower orders are connected with the work of vocal cords. Those of higher orders are related to changes in the human vocal tract (geometric, mass, form) [61].

**Hypothesis** **7.**When a subject is depressed, their tonal range decreases, so they tend to speak lower, flatter, and softer. Therefore, it can be assumed that MFCCs would be a good measure for diagnosing depression. Additionally, MFCCs are popular features used in speech recognition systems, and their properties have been used in research on emotion recognition, depression, and speech disorders.

## 4. Results

This section reports experimental results of selection and classification.

### 4.1. Feature Selection

In our experiment, we used a high-dimensional feature set, which initially consisted of 288 descriptors. All the features were calculated using the Essentia C++ library [62]. Then, we juxtaposed several different selection methods to minimize the feature set: three wrapper methods (Sequential Forward Selection SFS, Sequential Backward Selection SBS [63], and Recursive Feature Elimination RFE [64]), two filter methods (Mutual Information MI [65] and Information Gain Ratio IGR [66]), and one embedded method (Lasso [67]). The methods were preceded by feature correlation analysis, which rejected features with a degree of correlation higher than 0.8. Through this operation, we reduced the initial set size from 288 to 127. The similarity of feature subsets is presented in Figure 5. The type of the most frequently selected speech signal features is presented in Figure 6.

Analyzing Figure 5, it can be seen that in most cases the similarity of feature sets is high, reaching about 60–70%. An outlier is MI and IGR methods (both are filter methods), where the similarity of feature sets to other methods is in the range of 25–35%. As can be seen in Figure 6, features related to MFCC coefficients comprise the majority, regardless of the type of selection method, and account for 50–60% of the total. However, in the case of prosodies and features related to LPC coefficients, it varies from method to method. Thus, the prosody dominates the LPC coefficient-related features for MI, LASSO, RFE, and IGR, while the opposite is true for SFS and SBS. The following subsection describes how this affects the classification results.

### 4.2. Classification

In the study, we employed five different classifiers taking into account various features selection methods. For each classification method we analyzed three different cases. The first attempt used the straightway CGI scale, where the classes and labels are as follows: 1—mania, 2—hypomania, 3—euthymia (stable state), 4—subdepression, 5—depression, 6—severe depression, 7—mixed state). The second attempt merged elevated mood (mania and hypomania) into mania and depressed symptoms (subdepression, depression, and severe depression) into depression. The remaining states from CGI (mixed and euthymia) are unchanged. The last attempt consisted of two classes: mentally healthy (where we used only euthymia state) and mentally ill (where all remaining states with decreased or raised mood were merged together). The number of examples in a given CGI category for the above-mentioned cases is presented in Figure 7.

The number of patients contributing to each category differed as well as the number of state changes per patient. Those differences were minimized by the study inclusion criteria related to the frequency of episodes as described in the methodology section. Tests were performed using 10-fold cross-validation. The results are presented in Table 2 for Random Forest RF [68], Table 3 for k-NN [69], Table 4 for AdaBoost, Table 5 for Logistic Regression LR [70], and Table 6 for multilayer perceptron MLP [71]. The number of features in each set was selected experimentally, and only the results achieved with the most effective feature sets are presented. The relationship between the number of features and the classification results is presented in Figure 8.

In the case of two states recognition (mentally healthy vs. mentally ill), the best results were obtained using MLP, and the feature selection method had little effect on classification quality. For three selection methods, namely MI, FFS, and FGD, the results are at the same level: 95%. The relationship between the number of features and the classification results obtained using the IGR selection method for two classes is presented in Figure 8 (first from the left). A confusion matrix for MLP and IGR selection for two classes is presented in Figure 9.

Analyzing the results of four states recognition, one can quickly notice that the best results were obtained using RF, and similarly, as in the previous case, the feature selection method had little effect on classification quality. The results were at the same level for two selection methods, namely MI and LASSO: 77.1%. Slightly lower results, at 75.7%, were received for MLP. The relationship between the number of features and the classification results obtained using the IGR selection method for four classes is presented in Figure 8 (in the middle). A confusion matrix for RF and MI selection for four classes is presented in Figure 10.

As one would expect, with division into seven classes relative to the CGI, the results are the lowest, although not significantly different from those obtained for four classes: 73.4% using RF with LASSO, 71.6% using MLP with IGR. The relationship between the number of features and the classification results obtained using the LASSO selection method for seven classes is presented in Figure 8 (first from the right). A confusion matrix for RF and LASSO selection for seven classes is presented in Figure 11.

## 5. Discussion and Conclusions

This paper investigated the possibility of classifying the level of mania or depression of unipolar and bipolar disorder patients using a Polish audio dataset. We comprehensively analyzed speech signal features to predict mania and depression levels according to the CGI scale. The results indicate that the recognition of seven classes of illness is possible with 73% accuracy while reducing the number of classes to four increases classification results by only 3%, which is statistically significantly higher compared to a chance level. Additionally, we were able to prove that it is possible to discriminate between healthy and mentally ill patients’ speech with 95% accuracy.

The accuracy results obtained for both seven and four classes are not sufficient to implement the proposed system as a reliable decision support in real-world clinical applications. However, this limitation can be partially attributed to the small size of the training dataset, especially in cases of hypomania and severe depression. The training set contained only 15 samples of hypomania and 36 samples of severe depression, which is inadequate for achieving a high level of generalization and certainty. However, the system proves very reliable in distinguishing between mentally ill and mentally healthy patients, which can aid clinicians in detecting changes in a patient’s condition. Furthermore, the dataset was collected in a real-life setting, providing a high level of ecological validity. The recordings were captured in an authentic application with background noise, making this database particularly valuable. It is worth noting that there is a considerable amount of missing information, as not all patients recorded their speech on a daily basis. Additionally, medical visits occurred at least once a month, which could impact the accuracy of the diagnosis since the patient’s condition was only assessed during these visits. In situations where the mental state of a patient, as evaluated by the CGI scale, remained unchanged between two consecutive medical visits, the status label assigned during the visit was applied to all the days between these visits. However, this might not have accurately represented the patient’s actual condition during those intermediate days.

The successful implementation of machine learning methods for mood disorder identification and classification holds promise in detecting uni/bipolar disorders and quantifying treatment response at an earlier stage. Utilizing additional information to complement speech features has the potential to enhance the accurate discrimination of the uni/bipolarity in the CGI scale. By increasing classification performance, such monitoring systems can be utilized as predictors of treatment response. Consequently, the entire framework can be deployed as a decision support system for psychiatrists.

## Figures and Tables

**Figure 1 sensors-24-04721-f001:**
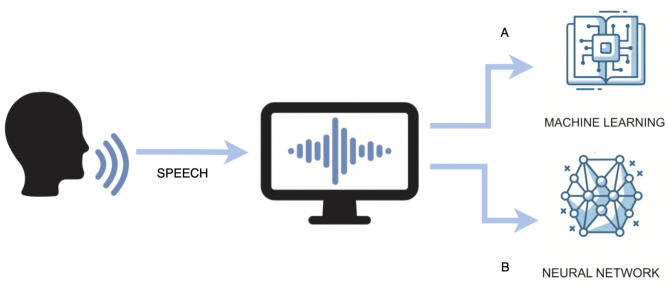
Typical speech recognition pipeline, the two-track approach. (**A**) Machine learning, (**B**) deep learning.

**Figure 2 sensors-24-04721-f002:**
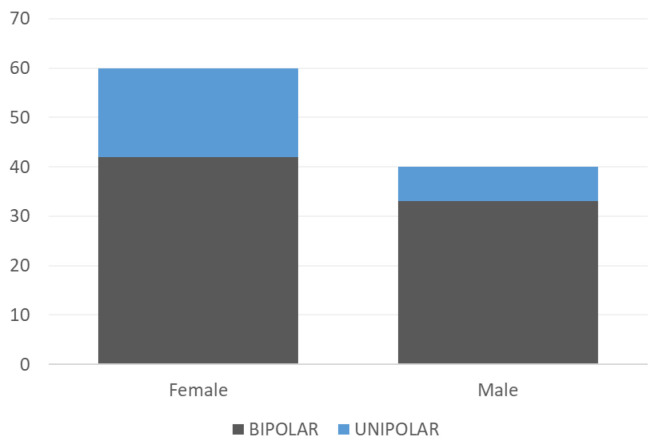
Distribution of the number of patients by gender and disease entity.

**Figure 3 sensors-24-04721-f003:**
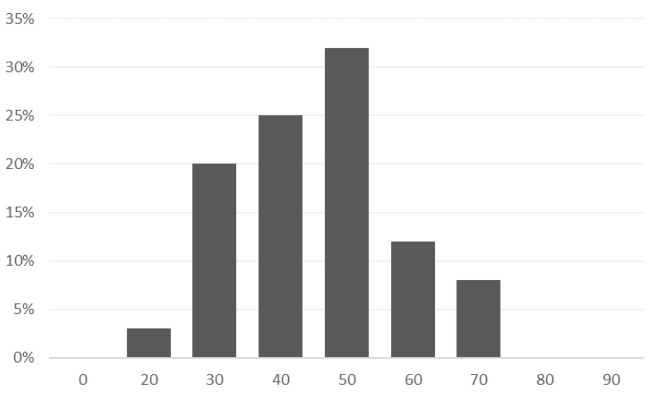
Distribution of the patients’ population by age.

**Figure 4 sensors-24-04721-f004:**
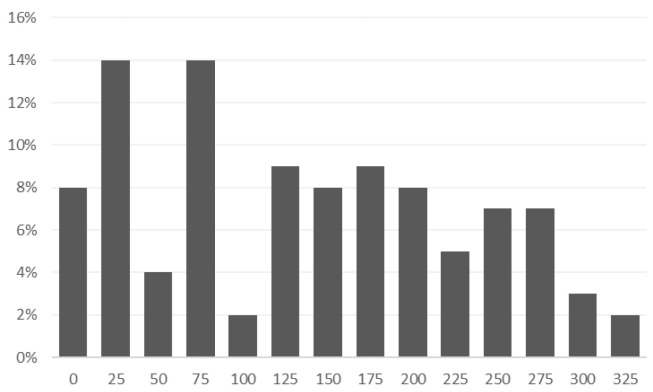
Histogram of the number of patient voice recordings.

**Figure 5 sensors-24-04721-f005:**
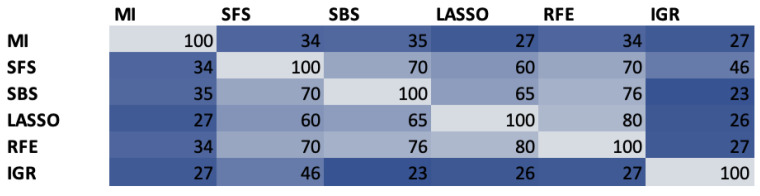
Similarity of features subsets selected using different techniques—example analysis for 100-feature set.

**Figure 6 sensors-24-04721-f006:**
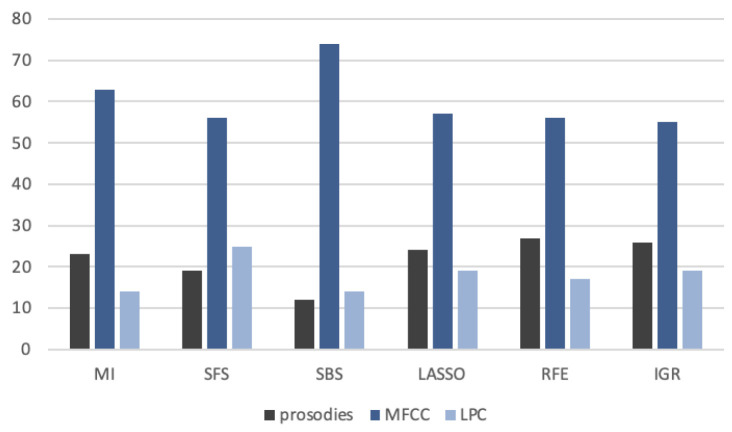
Type of speech signal features most frequently selected by different techniques—example analysis for 100-feature set.

**Figure 7 sensors-24-04721-f007:**
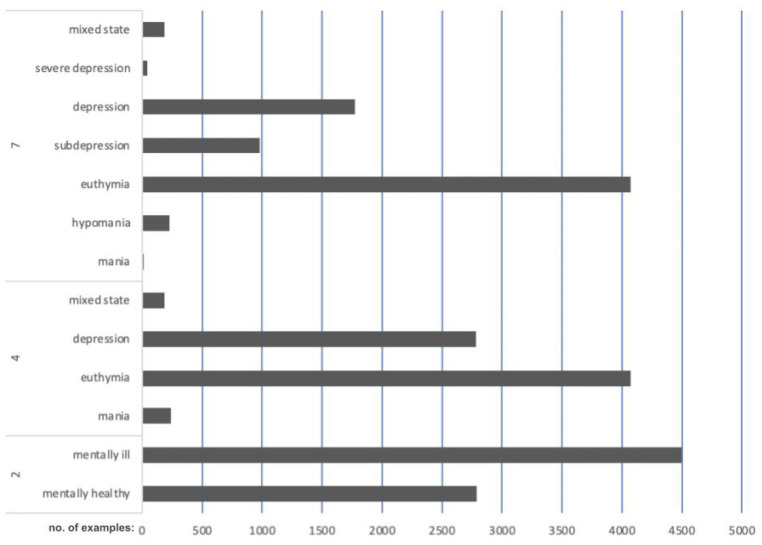
Number of examples in a given CGI category for three analyzed cases.

**Figure 8 sensors-24-04721-f008:**
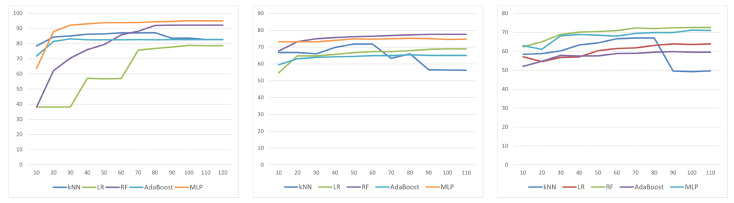
The relationship between the number of features and the classification results. IGR selection for (from left) 2, 4, and 7 classes.

**Figure 9 sensors-24-04721-f009:**
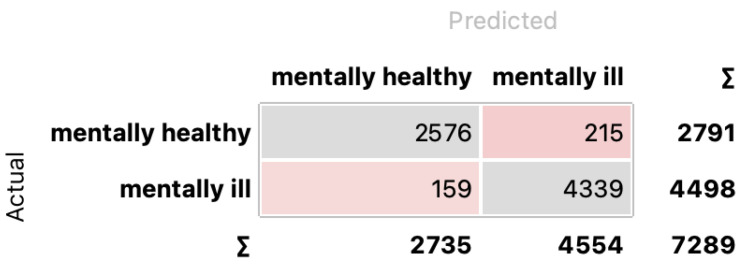
Confusion matrix for ML and IGR selection for two classes. In gray are highlighted correctly classified instances.

**Figure 10 sensors-24-04721-f010:**
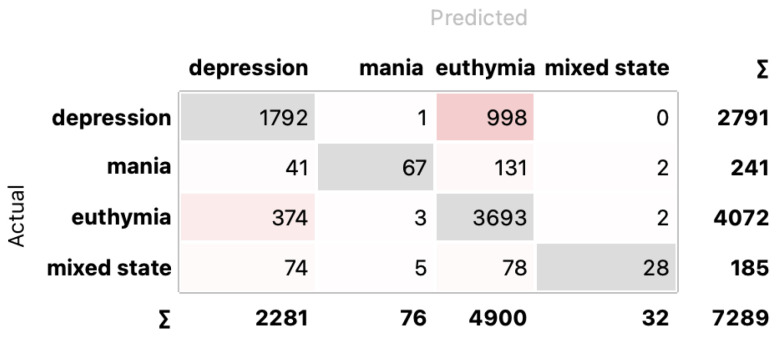
Confusion matrix for RF and MI selection for four classes. In gray are highlighted correctly classified instances.

**Figure 11 sensors-24-04721-f011:**
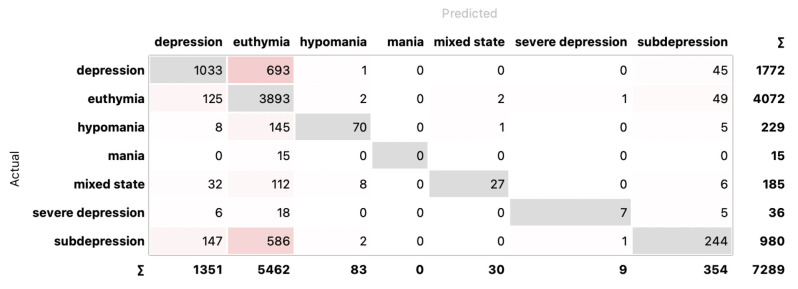
Confusion matrix for RF and LASSO selection for seven classes. In gray are highlighted correctly classified instances.

**Table 1 sensors-24-04721-t001:** Selected Datasets in Speech Depression Recognition.

Dataset	# of Subjects	# of Clips	Recording Method
AVEC2014 [22]	84 patients	300	reading aloud/questions responding
DAIC-WOZ [23]	189 patients	189	Wizard-of-Oz and automated agent
E-DAIC [24]	351 patients	275	Wizard-of-Oz and automated agent
MODMA [25]	23 depressed 29 control	1508	interviewing, reading, and picture description

**Table 2 sensors-24-04721-t002:** Classification results (average classification accuracy/F1 score in %) obtained with RF.

CGI#	MI	SFS	SBS	LASSO	RFE	IGR
7	71.7/67.7	72.6/68.5	72.6/68.5	**73.4/73.1**	72.7/68.3	72.2/68.4
4	**77.1/75.3**	76.3/74.2	76.9/74.8	**77.1/75.3**	75.9/73.9	75.9/73.9
2	**93.1/93.1**	92.8/92.8	92.4 /92.4	92.6/92.6	93.0/93.0	91.5/91.5

**Table 3 sensors-24-04721-t003:** Classification results (average classification accuracy/F1 score in %) obtained with k-NN.

CGI#	MI	SFS	SBS	LASSO	RFE	IGR
7	**66.2/65.4**	64.7/64.2	66.6/65.9	57.3/58.4	65.2/64.5	67.1/65.5
4	68.2/67.6	66.2/65.8	69.2/68.7	63.8/63.3	70.1/68.7	**71.8/69.5**
2	**89.7/89.7**	86.8/86.8	87.4/87.4	88.0/88.0	88.6/88.6	87.3/87.3

**Table 4 sensors-24-04721-t004:** Classification results (average classification accuracy/F1 score in %) obtained with AdaBoost.

CGI#	MI	SFS	SBS	LASSO	RFE	IGR
7	57.8/57.6	57.7/57.3	57.5/57.3	57.3/58.4	57.6/57.4	**59.5/59.3**
4	**66.3/64.5**	64.8/64.7	64.2/64.2	64.6/64.5	64.4/64.3	65.4/64.6
2	**83.1/83.1**	81.8/81.8	82.6/82.6	81.4 /81.4	81.6/81.6	82.8/82.8

**Table 5 sensors-24-04721-t005:** Classification results (average classification accuracy/F1 score in %) obtained with LR.

CGI#	MI	SFS	SBS	LASSO	RFE	IGR
7	**65.6/64.9**	62.9/62.7	69.2/59.9	62.3/62.4	63.6/63.3	63.8/63.4
4	66.8/66.2	68.5/67.8	67.1 /66.5	68.5/67.8	69.0/68.2	**70.3/69.2**
2	76.4/76.4	72.6/72.5	73.9/73.9	73.8 /73.5	74.7/74.4	**78.8/78.8**

**Table 6 sensors-24-04721-t006:** Classification results (average classification accuracy/F1 score in %) obtained with MLP.

CGI#	MI	SFS	SBS	LASSO	RFE	IGR
7	70.5/66.6	68.7/65.1	70.0/66.4	70.0/65.8	69.6/65.8	**71.6/67.6**
4	75.5/73.4	75.0/73.8	73.6/73.6	74.9/72.8	**75.7/73.8**	75.0/72.8
2	**95.0/95.0**	**95.0/95.0**	93.8/93.8	94.0 /94.0	94.8/94.8	**95.0/95.0**

## Data Availability

Data is unavailable due to privacy and ethical restrictions.

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
