# Peer review of "The Role of Selected Speech Signal Characteristics in Discriminating Unipolar and Bipolar Disorders"

_sensors, 2024, doi:10.3390/s24144721_

Round 1

Reviewer 1 Report

Comments and Suggestions for Authors

Although the impact and the relevance of the work is high, although the creation of a specific database of depression and bipolar disorder is important (and would deserve an improved description) this work shows several problems:

- experimental setup is not clear

- the measure for classification performance, i.e. accuracy, is misleading with such class size distributions

- the quality of acquisition is not evaluated

- the labeling procedure adopted between two successive visits does not guarantee a correct classification of the samples

- the state of the art related to observation of speech features taken into account is not complete

- the description of feature selection result is poor

In the methodology section, a more specific description of speech acquisition should be given

- which instructions were given to the subjects

- is there a way to suggest the user a way to perform acquisition in order to minimize unwanted environmental noise?

- which signal acquisition settings were adopted (sampling, coding, compression)?

- indicate duration of speech acquisitions?

- indicate the overall number of questions that were prepared by the psychiatry and specify whether the same question could be selected across different recording sessions? Please in

-list how many recording sessions were acquired for each subject, and the lengths of the recordings

-describe the acquisition of healthy controls; were the questions the same? how is it possible to disentangle possible biases due to the experimental paradigms (e.g. question related to disease to a healthy subject could elicit a specific kind of behavioural response). More details about the task and a discussion of possible biases should be given

- did the authors evaluate possible effect on speech feature of ANC of currently available smartphones? (authors are encouraged to cite the literature related to smartphone acquisition that is specifically focused on feature quality: several features could be influenced by ANC algorithms)

-  is there any possibility of checking recorded speech audio quality by the researchers? Authors should check audio quality and provide some criteria for the evaluation of acquired speech feature robustness. Some features such as jitter, are very difficult to be estimated in echological scenarios.

in the 4 classes scenario, hypomania and mania are merged in the same "mania" class? Please, describe the merging operations performed passing from the 7 class and the 4 class  scenarios.

The method section should describe also feature selection procedure and classifier; the result section should just focus on the results.

Authors should adopt a conservative approach and limit the analysis to the speech samples that are close to the clinical evaluation since the labeling of subject state for the intermediate days is likely to be wrong.

Authors should specify that no features from the same subjects are present both in the training and in the testing phases; it should be clearly stated and demonstrated from the results that training and testing datasets are independent.

As it was shown in the confusion matrix and discussed, some states are really represented by small numbers with respect to other states, and this should be taken into account, both in the choice of performance measures and in the interpretation. Results are shown in terms of accuracy, that when the size of the classes is heterogeneous, strongly depends on the more populated classes. Authors should consider using also F1 score and MCC. 

We are aware that mixed state classification represent one very difficult, but also relevant challenge: please consider that this might also be related to the fact that mixed state could be distinguished between mixed depression and mixed mania.

Discussion 

If the goal of the paper is to predict mania, this could not be accomplished.

the authors state that "The results indicate that recognition of seven classes of illness is possible with  73% accuracy while reducing the number of classes to four increases classification results by  only 3%, which is statistically significantly higher compared to chance level." The accuracy in this scenario, with such different sizes of the different classes, is misleading: its value is related to the results on the more populated classes. Also looking at the contingency table it is not possible to distinguish different classes

Also the sentence "Additionally, we were able to prove that it is possible to discriminate between healthy and mentally ill patient’s speech with the 95% accuracy." Given the lack of information about the tasks for the involved subjects, we cannot be sure that this claim is correct.

In the discussion "The training set contains only 15 samples of hypomania" : maybe the authors intended mania samples

Methods

When talking about  speech feature categories such as formants, f0, mfcc, jitter, please give specific indications on possible trends observed in depression and bipolar disorders

with reference to the literature

Fundamental frequency: in this paragraph depression is referred as emotion. Please, use the distinction between emotions and mood. Also, add some references about possible deviation of a general trend of lowered F0 in depression. For instance, there are several observations in the literature describing opposing trends of F0 with higher depression (see for instance the difference between agitate and retardated depression)

3.2.6. LPC and formants: the vocal tract should be indicated as the majour responsible of resonances in the speech production

results

The correlation among the features should be shown and discussed considering all the available features.

In figure 7 the number of examples in each conditions are shown. Are there any big differences across subjects regarding the number of examples in each state? (a subject could be observed in a given state much more frequently with respect another subject)

 The number of subjects contributing to each state should be shown as well as an index describing the mean and standard deviation of the number of examples in that given state across the observed population.

Axis labels are needed in Fig. 8

the features that were more frequently selected should be described

General comments

1) I would make a distinction between emotions and mood. However, i agree that speech features that  are used in emotion recognition can be used also for mood. I suggest the authors to discuss whether the time dynamics of the observed changes are relevant to distinguish between 

2) authors state "with the use of artificial intelligence can be a technological breakthrough in the diagnosis and effective treatment of affective disorders in the future."

why artificial intelligence could enhance diagnosis and treatment? 

Can artificial intelligence be used to enhance feature selection and classification performnce, or optimize the information in the data? 

Can AI alleviate the difficulties related to the phenomena under study, the social display problem, the search of relevant, behavioural, psychological and speech/voice features, the quality of acquired data?

Comments on the Quality of English Language

line 24 "mood disorders, not specified"; review sentence

line 175 posodies 

line 187 

Moreover a patient were examined by psychiatrist with regular visits least once a month where the BD mental state were assessed with CGI scale. review the sentence

4.1. Features selection  use instead 4.1. Feature selection

4.2. Clasification  use instead Classification

Author Response

Thank you for taking the time to review our article. We appreciate your valuable feedback and insights, which will help us improve the quality of our work. Our responses and comments are included in the attached file. Corrections to the article are highlighted in blue.

Reviewer 2 Report

Comments and Suggestions for Authors

To explore and enhance the diagnostic process of unipolar and bipolar disorders, this paper introduces a speech signal pipeline for CGI state classification, with a specific focus on selecting the most discriminative features. The results demonstrate a classification accuracy of 95% for the two-class classification, and 77.3% and 73% for the four- and seven-class classifications. However, there are some problems that should be considered as follows.

1.Most of the references are published before five years, please pay more attention to the latest studies.

2.Some key equations of feature extraction could be added in 3.2.2 to 3.2.7.

3. The collection method of speech signals in this study is not clearly described.

4. In the section of Discussion and Conclusion, this study should state and provide an association of the results with the literature and the results reported by previous research.

Author Response

The paper has been improved according to the reviews. Corrected sections are higlited in blue colour.

Methods description:

Given that speech features, selection and classification methods used in this research are extensively documented in the literature, we have opted not to provide detailed descriptions of these methods, while still adhering to proper citation practices. This approach is commonly employed.